# A Narrative Review on the Unexplored Potential of Colostrum as a Preventative Treatment and Therapy for Diarrhea in Neonatal Dairy Calves

**DOI:** 10.3390/ani11082221

**Published:** 2021-07-28

**Authors:** Havelah S. M. Carter, David L. Renaud, Michael A. Steele, Amanda J. Fischer-Tlustos, Joao H. C. Costa

**Affiliations:** 1Department of Population Medicine, University of Guelph, Guelph, ON N1G 2W1, Canada; havelah@uoguelph.ca; 2Department of Animal Biosciences, University of Guelph, Guelph, ON N1G 2W1, Canada; masteele@uoguelph.ca (M.A.S.); amandaf@uoguelph.ca (A.J.F.-T.); 3Department of Animal and Food Sciences, University of Kentucky, Lexington, KY 40506, USA; costa@uky.edu

**Keywords:** calf, diarrhea, colostrum, preventative, therapy

## Abstract

**Simple Summary:**

The rate of death and disease in young dairy calves is alarmingly high, with diarrhea accounting for over half of all disease incidence. The majority of diarrhea cases are treated with antimicrobials, which may not be warranted. There is increasing concern related to the use of antimicrobials in food animals due to the emanant development of antimicrobial resistance. Bovine colostrum is densely packed with hundreds of nutrients and bioactives specifically tailored to improve calf health and development. As such, colostrum may have potential to be used an alternative to antimicrobials for the treatment of diarrhea. The objective of this review is to outline the causation and implication of diarrhea in young dairy calves and to explore the benefits of colostrum and its potential use as a therapy and preventative for diarrhea in pre-weaned calves.

**Abstract:**

Diarrhea is the leading cause of morbidity and mortality in pre-weaned dairy calves and, as such, represents a significant animal health and welfare concern. Furthermore, digestive disease early in life is associated with several long-term consequences such as reduced growth rate and decreased milk yield during the first lactation, thus generating severe economic losses. The majority of diarrheic cases in young calves are treated with antimicrobials; however, it is necessary to develop alternative treatments, as excessive antimicrobial usage can lead to antimicrobial resistance and can negatively impact the gut microflora of a calf. Bovine colostrum is abundant in immune and bioactive factors that improve immune function and development. This rich and natural combination of immunoglobulins, natural antimicrobial factors, growth factors, anti-inflammatories and nutrients may be an attractive alternative to antimicrobials in the treatment of diarrhea in young dairy calves. There is evidence that supports the use of colostrum as an early treatment for diarrhea in young calves. Future research should investigate its therapeutic and economic effectiveness.

## 1. Introduction

High levels of disease in dairy calves threaten the economic efficiency and sustainability of the global dairy industry. Recent reports state that digestive diseases (i.e., diarrhea) are the leading cause of disease and death in the pre-weaning period [1]. Specifically, digestive disease accounts for 56% and 32% of morbidity and mortality, respectively [1], with the highest incidence in the second week of life [1,2,3]. Diarrhea is a multifactorial disease that results from a combination of exposure to pathogens, the environment and the host’s immune system [4]. The most common diarrheic pathogens include rotavirus and coronavirus, *Cryptosporidium parvum* (*C. parvum*), *Enterotoxigenic Escherichia coli* (*E. coli*), *Clostridium perfringens* and *Salmonella* spp. [3,5,6], most of which are highly contagious [7]. Exposure to one or more of these pathogens may result in diarrhea, which can be further exacerbated by a compromised immune system that may be a result of other diseases and poor management [8]. It is clear that the high incidence of diarrhea in young calves poses a significant concern from both welfare and health standpoints. Yet, diarrhea in pre-weaned calves can also have long-lasting economic effects [3], including a reduction in average daily gain (ADG) [9,10,11], an increased number of inseminations to achieve first pregnancy and an over 300 kg decrease in milk yield during the first lactation [11]. Consequently, research focusing on diarrhea therapy and prevention in young dairy calves is vital to the profitability of the dairy industry.

The majority of gastrointestinal diseases in calves are treated with antimicrobials [7]. Specifically, a study conducted in the United States found that 33.9% of all dairy calves were administered antimicrobials, with 75% of dairy calves with diarrhea receiving antimicrobial therapy [1]. There are a number of production, societal and health concerns pertaining to the frequent and improper use of antimicrobials in food production animals, including an increased risk of antimicrobial resistance [12]. Furthermore, animals administered antimicrobials, particularly young animals, can be negatively impacted by alterations to the gut microbiota in early life [13]. The diversity and abundance of total commensal gut microbiota can be reduced alongside targeted pathogens, upsetting microbial balance and causing gastrointestinal tract (GIT) dysfunction [14,15]. Due to the potential detrimental effects of high rates of antimicrobial administration, efforts should be made to reduce their use. Thus, the investigation of efficient alternatives to antimicrobials is necessary to improve both calf health and welfare, as well as consumer trust of the dairy industry and its products.

Bovine colostrum is densely packed with bioactive factors (i.e., antibodies, hormones and growth factors) and nutrients, resulting in a composition that differs substantially from mature milk [16]. Colostrum’s beneficial qualities have been used for centuries to provide several species with supplemental nutrition and treatment for numerous illnesses, including diarrhea in humans [17,18,19,20,21,22,23,24,25,26]. More recently, the advantages of colostrum as a prophylactic and therapeutic in young calves has been explored [27,28,29,30]. This research exemplifies that beneficial bioactives within bovine colostrum may provide an effective therapeutic alternative to antimicrobials for treatment of diarrhea in calves. Most research in this field involves the use of hyperimmune colostrum which is created by exposing the dam to specific pathogens, thus enriching her colostrum with pathogen specific antibodies [17]. This form of therapy can be expensive and lacks efficiency compared to natural bovine colostrum [17,31]; however, evidence supporting the use of natural colostrum as a therapy for diseases in cattle is lacking. Thus, the objective of this narrative review is to evaluate the physiology and effects of neonatal gastrointestinal diseases in dairy calves and, importantly, explore the potential benefits of colostrum as a prophylactic and therapeutic for neonatal calf diarrhea.

## 2. Causation of Digestive Disorders in Neonatal Calves

The dairy industry has made tremendous efforts to improve animal housing, hygiene, nutrition and health. However, neonatal calf diarrhea is still a major concern due to its multifactorial nature. As mentioned previously, digestive disorders can be a consequence of the environment, immunosuppression, pathogenic factors or a combination of the three [32,33]. Common diarrhea-causing pathogens include rotavirus and coronavirus (viral), *C. parvum* (protozoa), *E. coli*, *Clostridium perfringens* and *Salmonella* spp. (bacteria) [3,5,34]. *E. coli*, *C. parvum* and rotavirus have been found to cause >70%, >55% and 80% of cases of enteric illnesses in calves, respectively [2,3,5,35]. In addition, recent studies have demonstrated an increased frequency of multi-pathogen illnesses, with rotavirus and *C. parvum* being the most common pathogen combination [4,34]. An Australian study, which tested approximately 600 calves with diarrhea from 84 dairy and dairy beef farms, found that 96.4% of farms and 71% of samples contained multiple pathogens. In the infected calves, 42.8, 39.5 and 35.5% of samples contained two, three and four pathogens, respectively [5]. While pathogens can have synergistic and antagonistic effects on one another [3], evidence suggests that the difference between single pathogen and multi-pathogen diarrhea could be subclinical and clinical, respectively [4].

Although pathogens are associated with diarrhea, they can also be present in healthy calves. It has been determined that bovine coronavirus and rotavirus, as well as *C. parvum,* can be present in a calf without causing severe diarrhea [36]. Other factors, including failed transfer of passive immunity, poor nutrition, unhygienic conditions, hostile environmental conditions and poor management, can decrease immune function, allowing pathogens to thrive and ultimately cause neonatal calf diarrhea [5,6,8,35]. Specifically, routine and frequent cleaning of calf and maternity pens significantly decreases the risk of diarrhea [37]. Additionally, limiting other illnesses such as respiratory disease will reduce the risk of diarrhea caused by invasion of opportunistic pathogens [3,37]. Due to the aggressive nature of many diarrhea-causing pathogens, symptoms can be long-lasting and result in long-term consequences.

Diarrhea is a result of decreased absorption and increased secretions of water and electrolytes in the gut. This process generally begins with the ingestion of a single or multiple pathogens, which initiates numerous pathogenic physiological pathways leading to severe dehydration [33,38,39]. Consequently, the loss of fluids can cause a multitude of health and behavioral issues, including aversion to suckling, anorexia, depression, weakness and, in severe cases, death [7,33,38].

Rotavirus, coronavirus and *C. parvum* similarly cause villous atrophy by targeting small intestinal crypts that house stem cells that stimulate enterocyte renewal, see Figure 1 [38,39,40]. While rotavirus replicates in the cytoplasm of the epithelial cells in the small intestinal villi, coronavirus is released through cell lysis once replication is completed in enterocytes [38]. Rotavirus causes destruction to mature cells in the villi and activates non-structural glycoprotein 4, an enterotoxin that stimulates an influx of calcium ions, resulting in enterocyte destruction [41]. The pathophysiology of coronavirus is similar to rotavirus; however, this enveloped virus uses a spike protein to bind and fuse to the membrane of the host enterocyte, leading to atrophy [38]. *C. parvum* replicates through sexual and asexual reproduction, which creates autoinfecting thin-walled oocytes. The invasion results in atrophy of columnar epithelial cells and a loss of microvilli [38]. *Salmonella* destroys enterocytes by invading the intestinal mucosal layer, and continues to multiply in the lymphoid tissues [42]. The bacteria will then circulate through the body from mesenteric lymph nodes, initiating systemic illness [38,43]. Enterotoxigenic *E. coli* is another bacterium that is a major cause of diarrhea in calves. These bacteria attach to the immature epithelium through fimbria adhesion, mainly in the distal portion of the small intestine due to a favorably low pH. Hypersecretory diarrhea and dehydration results from the release of excess electrolytes and water due to villous atrophy though laminar propria damage and the release of thermo-stable toxins see Figure 1 [38]. Additionally, after enterocyte invasion, *Clostridium perfringens* may cause changes to the structure of the intestinal cytoskeleton through shortening of the columnar epithelial cells and destruction of microvilli due to necrotizing enteritis [38]. If not monitored and treated promptly, any of these pathogens can cause severe dehydration, malnutrition, reduced growth and death.

### Consequences, Concerns and the Economic Loss of Neonatal Calf Diarrhea

Diarrhea in pre-weaned calves can have a direct economic impact on producers through treatment and labor costs. Specifically, Goodell et al. (2012) calculated that a case of diarrhea in a pre-weaned calf costs USD 56, due to the cost of increased labor and medication. Beyond these costs, pre-weaning calf diarrhea carries a 5.1% case fatality rate [44]. Diarrhea in the pre-weaning period can also have long lasting effects, as it negatively impacts ADG by 50 g/day [11,32], and results in almost a 10% reduction in milk yield during first lactation [11,45]. In addition, calves with diarrhea are 2.9 times more likely to achieve first calving after 30 months of life [46]—six months later than optimal calving age [47]. A later age at first calving can lead to decreased lifetime milk production and increased costs to rear replacement animals [45]. Additionally, late calving age is associated with a higher weight at first calving, causing considerable metabolic and production complications during the transition period and throughout lactation [48]. It is clear that the negative impacts of pre-weaning diarrhea can have not only short-term consequences on calf health, but long-term effects on cow physiology and milk production. Thus, it is important to investigate strategies that reduce the high incidence of diarrhea, as well as potential alternatives to antimicrobials when clinical diarrhea occurs.

Aside from its economic impacts, pre-weaning diarrhea poses a major threat to calf welfare. There are myriad negative welfare challenges associated with diarrhea for pre-weaned calves, such as ataxia, dehydration through loss of fluids and weakness [49]. Furthermore, due to its immunosuppressive nature, calves with diarrhea have a higher likelihood of contracting other illnesses, such as respiratory disease [38]. Additionally, several symptoms caused by diarrhea may result in unnecessary stress and pain caused by anorexia and dehydration, which may ultimately result in death [3,50]. Furthermore, certain pathogens associated with calf diarrhea (*Cryptosporidium,* rotavirus and *Salmonella)* can be harmful to human health due to their zoonotic potential, as well as their abilities to develop antimicrobial resistance which can be passed on when meat and milk is consumed [6,42,51,52,53,54].

## 3. Antimicrobial Usage and Its Role in Diarrhea Therapy

Amoxicillin, chlortetracycline, neomycin, oxytetracycline, streptomycin, sulfachloropyridazine, sulfamethazine and tetracycline are all currently labeled in the United States for the treatment of calf diarrhea [7]. According to Urie et al. (2018), the antimicrobials listed above are utilized as a therapy for approximately 75% of calves diagnosed with diarrhea during the pre-weaning period. Due to a heavy reliance on these antimicrobials, pathogens such as *E. coli* and *Salmonella* have developed resistance to antimicrobials [55,56]. In addition, antimicrobial exposure can weaken the host’s immune system, increasing the risk for opportunistic pathogens, such as coronavirus, to colonize the gut [3]. Antimicrobial therapy is meant to target specific pathogens; however, this tends to force the gut microbiota into dysbiosis [14,15] through negatively impacting similar indigenous gut microbiota that play an important role in digestion and immunity [13,57]. The modification and deterioration of the composition of the gut microflora can lead to long-lasting negative effects [58], including increased gut permeability that allows for increased invasion of numerous pathogens [59]. Recent research in mice has shown that altered gut microflora composition negatively affects macrophage functions by pushing them into a hyperactive state [55], resulting in T-cell dysfunction and, ultimately, increased susceptibility to disease [55]. The pre-weaning period is a vital stage in gut microbiota establishment [56]. Disruption of gut microbial homeostasis during this period can have detrimental effects on calf health, including increased sensitivity to enteric infection, systemic disease and autoimmune disorders [58]. Due to the negative implications of antimicrobial use on the host immune system and gut microbiome, as well as concerns relating to the emergence of antimicrobial resistance, there is a push for producers to reduce heavy reliance on antimicrobials and begin to source alternatives when treating and augmenting the production and growth of their animals in order to ensure social acceptance by the dairy industry [60].

## 4. The Role of the Gastrointestinal Microbiome in Maintaining Gut Homeostasis

An infant’s gut is sterile in utero, with gut microbiome colonization beginning at parturition and continuing through early life [61]. A similar process is suspected in the calf [61]; however, there is evidence that suggests microbiota development may begin prenatally [62]. Although diversity of the gut microflora is marginal at birth [63], during the first 48 h of life and beyond, the microbiome is highly variable and diversifies significantly [64]. With much of the gut microbiota colonization occurring through environmental interactions at and soon after birth [65], colostrum feeding in early life can also significantly influence and shape a calf’s microbiome due to its ample supply of energy sources which are provided to the gut bacteria [65,66]. Rectal microbiota samples from newborn calves at birth show composition of Firmicutes, Proteobacteria, Actinobacteria and Bacteroidetes (phyla) [64]. However, by the first day of life, significant changes occur and only Firmicutes and Bacteroidetes remain, which is mainly a result of the environmental microbiota and colostrum feeding [64]. In the pre-weaning period, the intestinal microbiome is mainly comprised of bacteria [67,68] with *Lactobacillus, Corynebacterium, Streptococcus, clostridium*, *Pepto streptococcus* and *Bifidobacterium* being among the 167 genera identified at one week of life in the intestinal tract [67,68]. The gut microbiome changes significantly and continuously as the calf develops, consumes more solid feed and is weaned off milk [67].

The gut microbiota plays an important role in stimulating the function of the host immune system. This is especially prominent in neonates, where the early interactions of microbial communities can initiate long term immunity through the development of the mucosal layer of the epithelial cell barrier [69,70,71]. The epithelial layer is necessary for many vital immune and digestive functions of the GIT, where it plays a role in nutrient absorption and facilitates microbial cross-talk, as well as acts as a barrier against bacterial invasion [13,71,72]. Bacteria in the gut have a plethora of other functions, including improving immune function, decreasing pH, mucosal gut barrier maintenance and increasing digestive capacity [73]. The gut itself provides a localized immune response by eliminating pathogens with antigen presenting cells, neutrophils and natural killer cells [74]. Additionally, a large proportion of T-cells are present in the GIT, which are critical in the defense against pathogens [74]. The colonization of commensal microbes and their subsequent effects on calf immunity during early life promote a healthy gut environment, which may provide protection against pathogens and the high incidence of diarrhea in young calves.

## 5. Colostrum

Colostrum, the first milk produced following parturition, functions to aid in promoting growth and development of the neonatal calf, but it also supplies maternal antibodies and cell-mediated immunity to stimulate immune function within the first hours of life See Table 1 [2]. On-farm colostrum management has the greatest association with calf morbidity and mortality [75]; thus, it is vital to monitor the quality, quantity and cleanliness of colostrum and to ensure the calf receives the first colostrum feeding in a timely manner [76,77]. To be characterized as “good-quality”, colostrum should contain > 50 g/L of immunoglobulin G (IgG) to ensure successful transfer of passive immunity [77]. Quality can be altered by many factors, such as breed and age of the dam, season of calving, length of previous lactation and delayed colostrum collection [77]. Additionally, nutrition and health during the periparturient period of the dam such as energy and length of the dry period have shown to affect colostrum IgG levels [78]. In terms of quantity, calves should receive 10–12% of their body weight in colostrum at their first feeding, which over 50% of Canadian producers achieve [77,79], and ensure consumption of at least 150–200 g of IgG during the first 24 h of life [77,80]. Due to gut closure occurring during the first day of life, it is vital that colostrum is fed soon after birth to supply the calf with ample immunoglobulins [76]. Additionally, the cleanliness of colostrum is crucial to calf vitality. The method in which colostrum is collected and stored can influence the calf’s metabolism, endocrine system and nutrition [19,81]. The majority of Canadian dairy farmers reported that they regularly remove calves from the calving pen within 30 min after birth and prohibit suckling from the dam [79]. If contamination with bacteria occurs during the collection process, bacteria can bind to immunoglobulins, inhibiting their transport across the enterocytes and thus reducing transfer of passive immunity [77].

Bovine colostrum is abundant in immunoglobulins, which are more than 100 times greater in colostrum than mature milk, to ensure protection against disease [16]. The common isotopes include immunoglobulin M (IgM), immunoglobulin A (IgA) and IgG, of which IgG is the primary isotope [2] accounting for 85–90% of all bovine colostral immunoglobulins [82]. Immunoglobulins are composed of short and long polypeptide chains, with each class and sub-class differing slightly in chain formation [82].

During the first 24 h of life, maternal immunoglobulins are absorbed through colostrum to provide passive immunity to the calf [76]. After ingestion, immunoglobulins transfer into the neonate’s circulatory system through the lumen of the small intestine [16], where the calf is provided with short-term, immediate immunity [83]. As gut permeability decreases rapidly during the first day of life [84], delaying colostrum intake to as late as 12 h of life will concurrently reduce passive transfer of immunoglobulins [85]. Prior to absorption, immunoglobulins will also defend against disease, binding to pathogens on antigen binding sites of the intestinal mucosal membrane and thus preventing further intrusion or adhesion to the epithelium through the provision of an intestinal barrier [17,86,87]. Colostrum also contains compounds that are responsible for the protection of immunoglobulins, such as trypsin inhibitor, which is 100 times more abundant in colostrum. This prevents degradation of IgG in the GIT, thus increasing the availability of IgG for the calf [77]. 

In addition to the supply of immunoglobulins in colostrum, the latter contains an abundance of antimicrobial components. Lactoferrin, a bioactive protein in colostrum, has been shown to prevent sepsis in infants and calves, which can occur in calves with diarrhea [88,89]. It exhibits antimicrobial characteristics by creating a localized iron deficiency in bacteria though its binding capabilities, thus minimizing the potential for bacterial growth [16]. Moreover, it inhibits the growth of many microbes, including *E. coli* and *Salmonella* [90,91,92,93]. Lactoperoxidase is a similar bioactive compound that exhibits antimicrobial qualities by inhibiting bacterial metabolism through the suppression of oxidation in protein groups [16]. Another bioactive compound, lysozyme, can actively protect the host from Gram-positive and Gram-negative strains of bacteria by hydrolyzing the β 1–4 linkages in the cell wall, thus causing cell lysis [16,94].

Colostrum is also rich in additional bioactive molecules [95,96], such as insulin, insulin-like growth factor-I (IGF-I) and IGF-II [96]. Following parturition, their concentrations decline rapidly; therefore, early collection of colostrum is vital to ensure that the concentrations of both IgG and bioactive molecules are not diluted by transition milk [96]. Insulin has been shown to improve the oral glucose absorption capabilities of the calf, thus increasing its electrolyte levels [97]. Growth factors have been shown to stimulate the growth and maturation of the intestines, as well as the other mammalian cells [16]. Specifically, epithelial cell growth and development is stimulated by IGF-I and IGF-II [16]. Similarly, colostrum is rich in additional bioactive factors related to the innate and acquired immune system that are useful in the treatment and prevention of certain illnesses [98]. These include neutrophils, macrophages, immune regulators and anti-inflammatory molecules [98]. Neutrophils and macrophages are responsible for eliminating pathogens directly through phagocytosis, as well as through the production of cytokines [98]. Cytokines, including IL-1β, IL-6, TNF-α and INF-γ, regulate immune function, infection and stress though immune cell recruitment, meditation of inflammatory processes, enhancing phagocytotic abilities and immunological maturation [18,99]. In addition to factors that provide direct antimicrobial support with endotoxin-neutralizing abilities, colostrum contains several more bioactive molecules such as leptin, casein and α-lactalbumin that can reduce gut inflammation to encourage tissue repair and improve gastrointestinal mucosal integrity [18,100].

Although the concentrations of colostral components can vary, colostrum typically contains less than half the amount of lactose and more than 70% of oligosaccharides found in whole milk [101]. High levels of sialylated oligosaccharides are beneficial, as they have many potential prophylactic functions [101,102]. During a disease state, oligosaccharides provide several benefits through the inhibition of the binding capacity of certain pathogens, including *E. coli* and rotavirus, to the intestinal epithelium [102,103,104,105,106] and through the reduction in gut permeability by enhancing the expression of lipoproteins that comprise the tight junctional complexes [105,106]. Furthermore, oligosaccharides also decrease intestinal inflammation by reducing the expression of pro-inflammatory cytokines, which modulate proteins in epithelial tight junctions and enhance the expression of anti-inflammatory cytokines [105,106]. In addition, certain oligosaccharides may accelerate and improve the development of gut microflora in the calf by acting as an energy source for bacteria [104,105,106,107].

Overall, the fat content of colostrum is higher than that of transition or mature milk [95]. Elevated concentrations of certain fatty acids in colostrum insinuates that lipids may be essential for neonatal calves [108]. Neonatal ruminants are born with low energy stores and a limited ability to thermoregulate [109], as only 3% of a calf’s body weight is made up of lipids at birth [110]. The high levels of fat in colostrum are necessary to provide energy and to improve the calf’s thermoregulatory abilities [109]. Moreover, parturition activates the calf’s metabolic and respiratory processes, exposing it to high levels of oxidative stress [111]. Although the majority of lipids are in the form of triglycerides [112], the increased concentration of polyunsaturated fatty acids in colostrum has been shown to reduce the numbers of oxidants and reactive oxygen and nitrogen species, providing a greater antioxidant capacity and decreasing oxidative stress experienced by the calf [111,113].

**Table 1 animals-11-02221-t001:** Benefits to gut immunity and development provided by colostrum components.

Colostrum Bioactive		Concentration			Benefit to the Gastrointestinal Tract	Reference
		Unit	Colostrum	Mature Milk		
Immunoglobulin G		g/L	81	<2	Primary immunity contributor through pathogen binding in the intestinal mucosal membrane and passive immunity when absorbed into the circulatory system.	[17,77,86,87,96]
Lactoferrin		g/L	1.84	0.1	Sepsis prevention in infants. Binds to iron, preventing excess growth of bacteria, such as *E. coli* and *Salmonella*.	[16,88,89,90,91,92,93,96,114]
Lactoperoxidase		g/L	0.011–0.045	0.013–0.030	Inhibitory effects on bacterial metabolism through suppression of oxidation in proteins.	[16,114]
Lysozyme		μg/L	140–700	70–600	Cell lysis caused by hydrolysis of β linkages in the cell wall of Gram-positive and Gram-negative bacteria.	[16,94,114]
Insulin		μg/L	65	1	Promotes cell growth in the small intestine.	[96,115]
Insulin-like growth factor-I		μg/L	310	<2	Stimulates intestinal cell growth and epithelial development.	[16,96]
Insulin-like growth factor-II		μg/L	150	1	Stimulates intestinal cell growth and epithelial development.	[16,96,116]
Oligosaccharides		g/L	1	<0.2	Reduces gut permeability and promotes gut microflora development.	[101,117]
Fatty Acids		g/L	64	39	Improves thermoregulation capabilities. High levels of PUFA decreases oxidative stress by reducing the oxidants and reactive oxygen and nitrogen species.	[96,107,109,111]
Cytokines	IL-1 β IL-6 TNF-α INF-γ	μg/L	845 75 925 260	3 <0.2 3 0.2	Anti-inflammatory capabilities through the neutralization of pro-inflammatory molecules. Specifically, INF-γ amplifies the capacity of phagocytic cells.	[99,118]

### 5.1. Hyperimmune Colostrum as a Treatment for Diarrhea

Hyperimmune colostrum has been used as a therapy for gastrointestinal diseases in the dairy industry for many years [17]. It is produced by exposing the dam to a specific pathogen, which will enrich her colostrum with IgG specialized to recognize and protect the neonate against the specific pathogen in question [17,119]. This antibody-enriched colostrum has been used in human infants, children and adults to treat diarrhea caused by coronavirus [21], rotavirus [23] and *C. parvum,* alleviating symptoms for up to 3 months at a time with a single dose [22,24,25]. Hyperimmune colostrum has also shown to be effective in reducing the concentration *C*. *parvum* in the GIT of mice [120] and decreasing calf mortality during a coronavirus challenge [21]. Treating gastrointestinal diseases with hyperimmune colostrum is an attractive alternative to antimicrobials, as there are no effects on the diversity of the gut microflora [121]. Although hyperimmune colostrum poses several benefits, it is not feasible on a mass scale for treatment of calf diarrhea due to cost and production time. The use of hyperimmune colostrum may also pose societal concerns from both cow welfare and health standpoints due to the purposive administration of pathogens to create a product for the animal production industry. Furthermore, as stated previously, calf diarrhea is a multifactorial disease, and a single case of diarrhea can be caused by multiple pathogens. Thus, it would prove increasingly difficult and expensive to expose cows to multiple pathogens to create a hyperimmune colostrum that is effective towards the abundance of pathogens implicated in calf diarrhea. When the inefficiency and potential welfare and health concerns of hyperimmune colostrum are considered, it is clear that natural colostrum may act as a more suitable option to prevent and treat calf diarrhea.

### 5.2. Natural Colostrum as a Preventative for Calf Disease

Failure to absorb an adequate quantity of high quality immunoglobulins, especially IgG, is termed failed transfer of passive immunity, which results in a deficient immune system and increased risk of disease [76]. A newly recommended scale has been designed to measure transfer of passive immunity in four categories: >20.5, 18.0–24.9, 10–17.9 and <10 IgG/L representing excellent, good, fair and poor, respectively [122]. Achievement of good and excellent rates of passive transfer are used as a metric to indicate a lower risk for, and less susceptibility to, morbidity and mortality [122]. Although correlations between morbidity and mortality and the newly recommended passive transfer categories have been clearly demonstrated, it is necessary to determine correlations between each category and the incidence of specific diseases, including both respiratory and digestive disease.

Recently, there have been several studies highlighting the benefit of prolonged supplementation of colostrum beyond the first day of life. Kargar et al. (2020) determined that the enhancement of five liters of milk replacer with 0.7 kg/day of colostrum for 14 days postnatally improved ADG and decreased the risk of elevated rectal temperatures, diarrhea and pneumonia. Likewise, Berge et al. (2009) concluded that enhancing milk replacer with 70 g of colostrum replacer for 14 days increased grain consumption and ADG while reducing diarrhea incidence and antimicrobial administration. In addition to improved health, transition milk has also been shown to improve the development of the gastrointestinal tract [19]. Feeding colostrum for 72 h postnatally has been shown to improve the development and maturation of the small intestine, as well as improve IgG persistency [123,124]. Similarly, an enhanced nutritional plane in the pre-weaning period results in improved mammary tissue growth and mammogenic stimulation [125]. However, it is difficult to distinguish whether the benefits of the majority of the aforementioned studies are a result of extra nutrients, bioactives or a combination of the two, as energy and protein provisions are rarely balanced between treatments. Yet, this collection of studies suggests that an elevated colostrum allowance following the first day of life improves calf health and development, thus acting as a short- and long-term prophylactic for disease. Specifically, this research, along with the current body of literature highlighting the beneficial effects of colostral bioactive molecules on calf GIT development and health, indicate the great potential for colostrum as a preventative treatment for the current high rates of calf diarrhea.

### 5.3. Natural Colostrum as a Treatment for Diarrhea

Studies evaluating the use of natural colostrum have also shown positive results when using it as a treatment for different forms of diseases in humans. Bovine colostrum has been shown to improve gut immunity following certain diseases or treatments, such as short term bowel syndrome and chronic pain syndrome [18]. It has been found to minimize the immunosuppressive effects of intense physical activity through enhanced recovery and improved immune response [126]. Moreover, owing to the antimicrobial and anti-inflammatory properties of bovine colostrum, it can be beneficial in treating many different infections [18].

Few studies have explored the efficacy of bovine colostrum as a treatment for disease in neonatal calves. Cantor et al. (2021) highlighted how the use of colostrum may reduce disease likelihood in calves that had shown early disease symptoms. This research exemplifies that colostrum has an advantageous influence on calves displaying signs of morbidity; however, further research must investigate the effects of colostrum on diarrhea specifically. Bovine colostrum holds the potential to become a sustainable and effective therapy for neonatal calf diarrhea [127]. Colostrum has shown to naturally improve development and microbial colonization and development, as well as minimize inflammation within the GIT [66,77]. Critically high levels of antimicrobial resistance, in addition to the physiological upset antimicrobials pose to a young calves’ GIT, warrant a shift in therapeutics used for the treatment of diarrhea in pre-weaning calves [14,55,56].

Feeding colostrum after gut closure may have residual benefits. Although absorption will have ceased, immunoglobulins can still provide protection against pathogens in the GIT [17]. Moreover, there is an abundance of components that can improve calf diarrhea such as oligosaccharides, lipids, growth factors and hormones antibodies that continue to protect and maintain a healthy GIT [77,128]. Furthermore, several bioactives in colostrum provide immunity through phagocytosis and the prevention of pathogen growth [77]. This abundant combination of nutrients and bioactives could be the resource that is necessary in the treatment of diarrhea in young calves.

## 6. Future Research Directions

There have been recent studies attempting to find alternative treatments for neonatal calf diarrhea, including essential oil, probiotics and prebiotics, most of which have shown to decrease calf mortality and improve ADG [129,130,131]. However, the current alternatives to antimicrobials consist of either a single treatment or a combination of 2–4 ingredients; contrarily, colostrum is a natural combination of hundreds of bioactives specifically tailored to an individual calf [77,129,130]. Knowledge is still lacking within this field and minimal research has been conducted to evaluate the impact natural bovine colostrum can have as a treatment for neonatal calf diarrhea. It is not known whether colostrum can lead to improved recovery from diarrheic symptoms or what the proper treatment protocol should be (i.e., required dosage and quality level of the colostrum). It is clear that an adequate quantity of colostrum fed quickly after birth will provide the calf with protection against the development of diarrhea. Additional research has highlighted that, even following gut closure, supplementation of colostrum can reduce diarrhea through its bioactive and immune enhancing components [29]. Other research has shown that, by adding colostrum to milk replacers, the probability that antimicrobial treatment would be required decreased by 57.7%, as colostrum acted as a protectant against diarrhea, respiratory disease and navel ill [30].

## 7. Conclusions

The dairy industry is making tremendous efforts to improve animal welfare and health through improved hygiene, management and environment. However, several diseases in pre-weaning calves still pose a serious issue. With antimicrobial resistance steadily growing, there has been a recent influx in research related to antimicrobial alternatives. Colostrum holds the potential to be used as a natural prophylactic and therapy for diarrhea in young calves. Using future research regarding the development of protocols, this therapy may be a practical and natural method to treat pre-weaning calves with diarrhea while adhering to industry and consumer standards.

## Figures and Tables

**Figure 1 animals-11-02221-f001:**
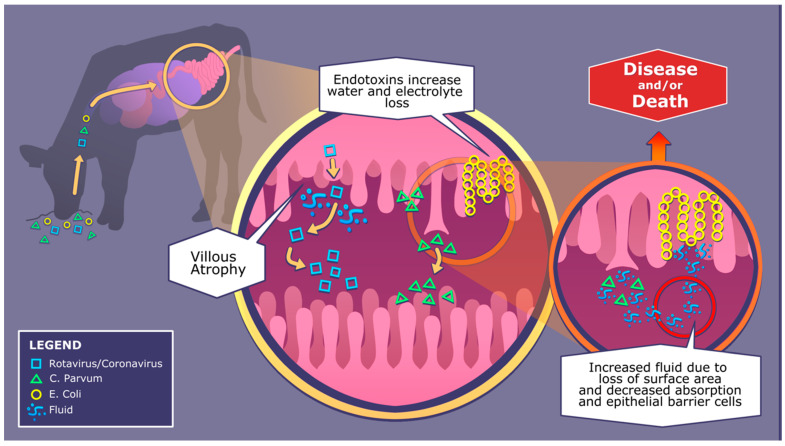
Pathophysiology of diarrhea-causing pathogens; rotavirus, coronavirus, *C. parvum* and *E. coli*.

## Data Availability

Data are contained within the article.

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
