# Peer review of "A Narrative Review on the Unexplored Potential of Colostrum as a Preventative Treatment and Therapy for Diarrhea in Neonatal Dairy Calves"

_animals, 2021, doi:10.3390/ani11082221_

Round 1
Reviewer 1 Report
Dear Authors, the manuscript is well-written and brings light into a serious issue often faced by dairy producers. I have some comments before considering the article for publication:
- Nutritional/health of pregnant dairy cows is a concern during the periparturient period, which coincide with colostrum production. Hence, it would be important to address these challenges that producers face regarding the status of the pregnant dairy cow in order to obtain a high-quality colostrum.
- Is there a survey in which colostrum feeding and quality was thoroughly evaluated? If so, this could be cited and discussed in the manuscript to present the current overview of these aspects
- The authors mention the effects of diarrhea on several production, health, and reproduction parameters. But is there a transgenerational effect as well? In other words, calves born from cows that suffered from diarrhea also present a case of diarrhea?
- Eimeria is also a big concern for dairy producers, occurrin between 3-4 weeks. Is there evidence that colostrum feeding can also benefit this kind of diarrhea?
- Are there any associative effects of enriching milk replacer with colostrum + probiotics? Some articles have been evaluating these molecules as possible alternatives to improve gut health and to reduce disease occurrence in dairy calves
Author Response
Reviewer 1: Nutritional/health of pregnant dairy cows is a concern during the periparturient period, which coincide with colostrum production. Hence, it would be important to address these challenges that producers face regarding the status of the pregnant dairy cow in order to obtain a high-quality colostrum.
AU: Thank you for your comment. I have added information into the manuscript. Lines 246-248
Reviewer 1: Is there a survey in which colostrum feeding and quality was thoroughly evaluated? If so, this could be cited and discussed in the manuscript to present the current overview of these aspects
AU: Thank you for your comment. This is a great addition. Lines 249-250 and 255-257
Reviewer 1: The authors mention the effects of diarrhea on several production, health, and reproduction parameters. But is there a transgenerational effect as well? In other words, calves born from cows that suffered from diarrhea also present a case of diarrhea?
AU: Thank you, I agree with your comment. This is an interesting question, but the literature is lacking in this area, so we did not focus on this topic. There is evidence that diarrhea will cause long-term effects with regards to health and production of that animal (Abuelo, et al. 2021). Similarly, it has been shown that stress on a dry cow and calf, specifically heat stress can have transgenerational effects (Ouellet, et al. 2020), however, little research has been done related to how cases of diarrhea affect the offspring.
Reviewer 1: Eimeria is also a big concern for dairy producers, occurring between 3-4 weeks. Is there evidence that colostrum feeding can also benefit this kind of diarrhea?
AU: I agree, this is a big concern. Several papers have been published regarding colostrum supplementation within the first two weeks of life such as Kargar, et al. 2020, Cantor, et al. 2020 and Berge, et al. 2009, however, research is lacking on the effects of colostrum as a treatment after the first two weeks of life as the majority of diarrhea occurs within the first 14 days.
Reviewer 1: Are there any associative effects of enriching milk replacer with colostrum + probiotics? Some articles have been evaluating these molecules as possible alternatives to improve gut health and to reduce disease occurrence in dairy calves
AU: Thank you for your comment. I have added details into the manuscript. Lines 433-434

Reviewer 2 Report
This article comprises a very complete and important review on the properties of bovine colostrum as a potential antimicrobial and its ability to be used in the prevention and treatment of calf diarrhea. Additionally, this article addresses the role of the livestock industry in the development of alternatives for antibiotic use in an effort to reduce bacterial resistance. The article is well written and addresses important epidemiological, etiological, and pathophysiological components of calf diarrhea and introduces the concept of maternal colostrum as an alternative to antimicrobials in the treatment and prevention of the disease in cattle operations. For these reasons my opinion is that this article is accepted for publication after minor revisions.
Specific comments to the authors:
-Line 41. Introduction. Clostridium perfringens are normal inhabitants of the GIT of cattle in their unsporulated form. Only when favorable conditions are present and sporulation occurs activating release of exotoxins such as alpha, beta, or epsilon is that severe necrotizing enteritis and other clinical syndromes are observed in affected calves. Additionally, only one or few calves are usually affected. Therefore, this makes of this pathogen an infectious but not highly contagious pathogen. Please revise and correct this sentence.
-Line 121. Please change "physiology" to "pathophysiology"
-Line 128. Enterotoxigenic E. coli (K99) causes a hypersecretory diarrhea due to production of thermo-stable toxins after adhesion of its fimbria (pili - K99) to immature enterocytes (crytps) during the first 3 days of life. These thermo-stable hypersecretory toxins increase secretion of electrolytes and water leading to diarrhea and dehydration. Please revise and correct.
-Line 132. Related with C. perfringens, please check previous comments and revise this sentence.
-Line 198. Do you mean "ample" instead of "apple"?
-Line 203. Please change "periosd" for "period"
-Lines 255-258. The sentence is too long and difficult to understand, please re-phrase.
-Lines 261-261. This is redundant as it had been mentioned before.
-Line 298. Do you mean "disease" instead of "diseased"?
-Lines 380-381. The explanatory objective of this sentence is difficult to understand. Colostrum decreases immune suppressive effects of high intensity physical exercise by decreasing innate immunity? please re-phrase and explain.
-Line 395. Please check the presence of "has occurred" in the sentence as it seems misplaced.
Author Response
Reviewer 2: Line 41. Introduction. Clostridium perfringens are normal inhabitants of the GIT of cattle in their unsporulated form. Only when favorable conditions are present and sporulation occurs activating release of exotoxins such as alpha, beta, or epsilon is that severe necrotizing enteritis and other clinical syndromes are observed in affected calves. Additionally, only one or few calves are usually affected. Therefore, this makes of this pathogen an infectious but not highly contagious pathogen. Please revise and correct this sentence.
AU: Thank you for you insight. I have changed "all" to "most" on line 41.
Reviewer 2: -Line 121. Please change "physiology" to "pathophysiology"
AU: Changed Physiology to pathophysiology on line 122.
Reviewer 2: Line 128. Enterotoxigenic E. coli (K99) causes a hypersecretory diarrhea due to production of thermo-stable toxins after adhesion of its fimbria (pili - K99) to immature enterocytes (crytps) during the first 3 days of life. These thermo-stable hypersecretory toxins increase secretion of electrolytes and water leading to diarrhea and dehydration. Please revise and correct.
AU: Thank you for your input. I have revised and corrected the manuscript on lines 130-134
Reviewer 2: line 132 Related with C. perfringens, please check previous comments and revise this sentence.
AU: Thank you for your insight. Lines 135-137 have been changed as per the information you have provided.
Reviewer 2: Line 198. Do you mean "ample" instead of "apple"?
AU: Thank you for catching this. Revised and corrected. Line 208
Reviewer 2: -Line 203. Please change "periosd" for "period"
AU: Thank you for catching this. Revised and corrected. Line 213
Reviewer 2: Lines 255-258. The sentence is too long and difficult to understand, please re-phrase.
AU: Thank you for your input. I have divided the sentence to improve fluidity. Lines 272-274
Reviewer 2: Lines 261-261. This is redundant as it had been mentioned before.
AU: I have removed this sentence from the manuscript. Thank you.
Reviewer 2: -Line 298. Do you mean "disease" instead of "diseased"?
AU: Revised and corrected “diseased” to “Disease”. Line 324
Reviewer 2: Lines 380-381. The explanatory objective of this sentence is difficult to understand. Colostrum decreases immune suppressive effects of high intensity physical exercise by decreasing innate immunity? please re-phrase and explain.
AU: Thank you. I have rephrased this sentence to make it more comprehensive. Line 408.
Reviewer 2: Line 395. Please check the presence of "has occurred" in the sentence as it seems misplaced.
AU: Thank you. Removed “Has occurred”. Line 423
